# Synthesis, Anti-Proliferative Evaluation and Mechanism of 4-Trifluoro Methoxy Proguanil Derivatives with Various Carbon Chain Length

**DOI:** 10.3390/molecules26195775

**Published:** 2021-09-24

**Authors:** Simeng Xu, Yufang Cao, Yu Luo, Di Xiao, Wei Wang, Zhiren Wang, Xiaoping Yang

**Affiliations:** 1Key Laboratory of Study and Discovery of Small Targeted Molecules of Hunan Province, Key Laboratory of Protein Chemistry and Developmental Biology of Fish of Ministry of Education, Department of Pharmacy, School of Medicine, Hunan Normal University, Changsha 410205, China; simeng.xu@hunnu.edu.cn (S.X.); yufang.cao@hunnu.edu.cn (Y.C.); dixiao@hunnu.edu.cn (D.X.); zhirenwang@foxmail.com (Z.W.); 2TCM and Ethnomedicine Innovation & Development International Laboratory, Innovative Material Medical Research Institute, School of Pharmacy, Hunan University of Chinese Medicine, Changsha 410208, China; luoyu1998111@163.com (Y.L.); wangwei402@hotmail.com (W.W.)

**Keywords:** biguanide, anti-cancer, trifluoromethoxy, AMPK

## Abstract

Among the known biguanide drugs, proguanil has the best antiproliferative activity. In contrast, newly synthesized biguanide derivatives containing fluorine atoms have excellent biological activity, among which trifluoromethoxy compounds show the strongest ability. Preliminary work in our laboratory exhibited that n-heptyl containing proguanil derivatives on one alkyl chain side have better biological activity than those with a shorter carbon chain. However, the relationship between the length of the carbon chain and the activity of the compounds is unknown. In this study, we synthesized 10 new trifluoromethoxy-containing proguanil derivatives with various carbon chain lengths. The phenyl side is fixed as the trifluoromethoxy group with change of carbon chain length in alkyl chain side. It was found that the anti-cancer abilities of **5C**–**8C** with n-pentyl to n-octyl groups was significantly better than that of proguanil in the five human cancer cell lines. The colony formation assay demonstrated that **6C**–**8C** at 0.5 to 1.0 μM significantly inhibited the colony formation of human cancer cell lines, much stronger than that of proguanil. Pharmacologically, **8C** activates AMPK, leading to inactivation of the mTOR/p70S6K/4EBP1 pathway. Thus, these novel compounds have a great potential for developing new anti-cancer candidates.

## 1. Introduction

According to the data of The World Health Organization’s International Agency for Research on Cancer, it is estimated that there were 19.29 million new cancer cases and 9.96 million cancer deaths worldwide in 2020 [1]. Thus, finding efficient and safe targeted anti-tumor drugs is still an arduous and urgent task. Recently, biguanides have attracted considerable attentions due to their excellent biologically activities. Clinically, buformin and metformin have been used to treat diabetes [2,3], and proguanil is used to treat malaria [4]. In contrast, the anti-proliferation activity of biguanides, especially metformin, has stimulated great interests but has not been used in clinic yet [5]. Furthermore, preclinical studies have shown that the effective concentration of metformin in inhibiting cancer cell proliferation is at millimolar level [6], which is difficult to achieve in vivo through routine oral administration route, making it difficult for metformin to be used as a clinical anti-tumor agent [7]. Moreover, studies have reported that proguanil has the strongest inhibitory effect in bladder cancer and colon cancer cell lines among biguanides [8]. Therefore, the structural optimization of proguanil has the great potential to find more effective anti-proliferative compounds. 

Fluorine-containing derivatives are reported to have excellent activity including anti-proliferation and anti-cancer capabilities [9,10]. Among the fluorine-containing substituents, trifluoromethoxy has a significant impact on enhancing biological activity, so it has been received widespread attention [11,12]. Compared with other fluorine-containing substituents, trifluoromethoxy is relatively inert and exhibits stronger stability under heating, acidic or alkaline conditions [13,14]. In recent years, chemical drugs containing trifluoromethoxy have been approved by the US Food and Drug Administration (FDA) for the treatment of tuberculosis (Pretomanid) and basal cell carcinoma (Sonidegib) (Figure 1). Pretomanid, approved by the US FDA, is used in combination with bedaquiline and linezolid to treat specific highly drug-resistant tuberculosis (TB) patients [15]. Sonidegib is approved by the FDA for use in adult patients with locally advanced basal cell carcinoma [16]. In the previous research of our group, it was found that when there is n-heptyl substitution on the alkyl side of fluorine-containing proguanils, these compounds exhibit better antiproliferative activity [17], but the relationship between alkyl chain length and antitumor activity is not clear. In this study, a trifluoromethoxy group was introduced at the para position of the phenyl group with alteration of the carbon chain length on the side of the biguanide group (Figure 2). It was found that the anti-proliferative activity of **5C**–**8C** (n-pentyl to n-octyl) was significantly improved compared to proguanil, and when the carbon chain length reached **9C** (n-nonyl), the anti-proliferative activity was significantly reduced, indicating there is an optimized carbon atom length. Furthermore, it is still unknown whether these new biguanide derivatives affect the molecular signal pathways or not. Thus, anti-tumor mechanism of representative biguanide derivative was further investigated.

## 2. Results

The target derivatives **2C**–**10C** and **12C** were synthesized according to the method as shown in Figure 1. The commercially available compound 4- (trifluoromethoxy) aniline was firstly reacted with sodium dicyandiamide at 80 °C to obtain the intermediate. It was then reacted with corresponding alkylamines in tetrahydrofuran at 40 °C until the intermediate compound was fully reacted. Finally, after HCl solution was added and stirred for 30 min, ethylenediaminetetraacetic acid (EDTA) solution was dropped into the reaction mixture and filtered to obtain target derivatives **2C**–**10C** and **12C**. The main purpose of this study is to investigate whether the anti-proliferative activity of biguanides is increased when one side of the fixed benzene ring is trifluoromethoxy and the other side of the carbon chain lengthens. We synthesized 10 new compounds and determined their IC_50_ (Table 1). Three bladder cancer cell lines and two ovarian cancer cell lines were exposed to specific concentrations of all compounds to determine their inhibitory effects on cell proliferation. Since the anti-proliferation ability of compound **5C**–**8C** was significantly better than that of proguanil, we chose **6C**, **7C** and **8C** to perform clone formation experiments in three bladder cancer cell lines. As shown Figure 3, Figure 4 and Figure 5, these compounds between 0.5 to 4.0 μM exhibited excellent activities in inhibiting clone formation in these cancer cell lines, similar to the results of MTT. 

Western blotting was used to evaluate the effect of compound **8C** on AMPK signaling pathway in T24, one of the most commonly used cell lines. As shown in Figure 6, compound **8C** up-regulates p-AMPK and has inhibitory effects on downstream p-mTOR, p-p70S6K and p-4EBP1. 

## 3. Discussion

There are four synthesis routes for preparation of proguanil derivatives [18], we choose the route with highest yield and experimental feasibility as shown in Figure 1. Ten new compounds were successfully obtained and named **2C**–**10C** and **12C**. The results of MTT demonstrated that IC_50_ values of **5C**–**8C** are significantly lower than that of proguanil. In contrast, the anti-cancer activities of the derivatives with a longer carbon chain than **8C** have decreased significantly, even less than that of proguanil, indicating that there is an optimized carbon chain length in biguanide derivatives. In view of two ovarian cancer cell lines, all compounds have low sensitivity in SKOV3 cell line with less sensitive than that in OVCAR3. Interestingly, the sensitivity of these compounds to three bladder cancer cell lines is similar. In other hand, the results of colony formation demonstrated that **6C**, **7C** and **8C** have strong anti-clone formation capacities at concentrations between 0.25–1.0 μM while proguanil has no inhibitory effect on colony formation at a concentration of 1.0 μM in our previous study [17], indicating that compounds **6C**, **7C** and **8C** have much better anti-clone formation activities than that of proguanil in T24 and UMUC3. Similar results were obtained in J82, as shown in the Appendix A. 

Since there is no significant difference in the anti-proliferative activity between **5C** to **8C** while compounds with longer carbon chain than 8C have poorer activity than that of proguanil, we selected the compound **8C** as a representative compound to investigate the pharmacological mechanism. Previously, our research group has proved that biguanide derivatives can exert anti-tumor effects through AMPK/mTOR pathway [19]. Adenosine 5’-monophosphate (AMP)-activated protein kinase (AMPK) is a key molecule in the regulation of bioenergy metabolism [20], and it is the core of the study of diabetes and other metabolic-related diseases [21,22]. It is expressed in various metabolic-related organs and can be activated by various stimuli of the body, including cell pressure, exercise and many hormones and substances that can affect cell metabolism [23,24,25]. Mammalian target of rapamycin (mTOR) is an important regulator of cell growth and proliferation [26]. A large number of studies have shown that the abnormal regulation of mTOR signaling pathway is closely related to cell proliferation [27,28,29]. Compound **8C** up-regulates p-AMPK and has inhibitory effects on downstream p-mTOR, p-p70S6K and p-4EBP1. Instead, previous studies have shown that proguanil has no effect on the AMPK/mTOR pathway at a concentration of 2.0 μM [17]. In this study, compound **8C** can significantly regulate the expression of AMPK/mTOR pathway-related proteins at a concentration of 2.0 μM, indicating a stronger effect of 8C on AMPK/mTOR pathway than that of proguanil. In summary, our data demonstrate that the optimized biguanide derivatives in this study can indeed inhibit cancer cell growth associated with the AMPK/mTOR signaling pathway with a more effective activity than that of proguanil.

## 4. Materials and Methods

### 4.1. Chemistry

All solvents and materials are commercial grade and do not require further purification unless otherwise specified. Proton nuclear magnetic resonance (^1^H-NMR) spectra and ^13^C NMR were recorded on a Brucker DRX spectrometer (600 MHz) in dimethyl sulfoxide (DMSO) solvent by using TMS as an internal standard. High resolution mass spectrometry (HRMS) analysis was performed on an Agilent 1290 HPLC-6540-Q-TOF instrument. The purity of the derivatives was determined by high-performance liquid chromatography (HPLC), conducted on an Agilent 1260 HPLC system with column (Agilent 5 TC-C18(2) 250 × 4.6 mm), and the samples were eluted with a 1:1 methanol/H_2_O mixture, at a flow rate of 1 mL/min. The melting point of the derivatives was obtained by a micro melting point apparatus XT3A.

#### 4.1.1. Method for Synthesizing Compound **2**

NaN(CN)_2_ (5 g, 56 mmol) was matched into solution in 43 mL water, under the condition of 80 °C to join 4-trifluoro methoxy aniline solution (7.32 g, 30 mmol for 4-trifluoro methoxy aniline, soluble in water and concentrated HCl 2.5 mL/mL), 80 °C, continuous reaction 1 h or so, in the reaction liquid solid precipitation gradually, until the TLC test reaction liquid contains no 4-trifluoro methoxy aniline, at the end of the reaction. 1-cyan-3-(4-(trifluoromethoxy) phenyl) guanidine was obtained by filtration and vacuum drying.

#### 4.1.2. Method for Synthesizing Compound **2C**–**10C**, **12C**

At room temperature, 1 g (4.1 mmol) compound 2 was stirred and added to 5 mL THF and 4 mL H_2_O, followed by 0.7 g (2.8 mmol) of copper sulfate pentahydrate and 16.4 mmol of alkylamine. It was heat ed to 40 °C and continued to be stirred. After checking for the presence of compound 2 in the TLC, the tetrahydrofuran was evaporated by vacuum distillation. After cooling the reactant to 25–30 °C, HCl solution (2 mL concentrated hydrochloric acid in 3 mL water) was added and stirred for 30 min. Then, dropped cooled ammoniacal EDTA solution (3 mL water, 1.5 mL ammonia water (25%) and 1.4 g EDTA disodium salt) into the reaction mixture. The temperature range was kept at 15–20 °C. After adding, the product was stirred at the same temperature for 30 min, filtered and separated, washed repeatedly with cold water, dried at 90–95 °C, and solid target product was obtained.

##### *N*-1-Ethyl-*N*-5-(4-trifluoro Methoxy) Proguanil (**2C**)

Yield: 55.7%. Mp: 193–195 °C. ^1^H NMR (600 MHz, DMSO-*d_6_*) δ 9.83 (s, 1H), 7.54–7.25 (m, 4H), 6.86 (s, 1H), 3.13 (q, *J* = 7.3 Hz, 2H), 2.75–2.71 (m, 1H), 1.54 (s, 1H), 0.86 (td, *J* = 6.0, 5.4, 3.3 Hz, 3H). ^13^C NMR (150 MHz, DMSO-*d_6_*) δ163.98, 150.59, 143.81, 137.93, 121.97, 121.46, 119.77, 31.20, 14.32. HRMS (ESI) (*m*/*z*) [M+H]^+^ calcd for C_11_H_15_F_3_N_5_O, 290.1229; found, 290.1218. Purity: 99.74% (by HPLC).

##### *N*-1-Propyl-*N*-5-(4-trifluoro Methoxy) Proguanil (**3C**)

Yield: 52.8%. Mp: 195–196 °C. ^1^H NMR (600 MHz, DMSO-*d_6_*) δ 8.16 (s, 1H), 7.58–7.19 (m, 5H), 3.04 (t, *J* = 7.2 Hz, 2H), 1.46 (s, 2H), 0.92–0.79 (m, 3H). ^13^C NMR (150 MHz, DMSO-*d_6_*) δ 161.62, 155.91, 143.61, 138.96, 123.16, 122.23, 121.64, 119.77, 118.08, 40.51, 22.99, 11.69. HRMS (ESI) (*m*/*z*) [M+H]^+^ calcd for C_12_H_17_F_3_N_5_O, 304.1385; found, 304.1396. Purity: 97.75% (by HPLC).

##### *N*-1-Butyl-*N*-5-(4-trifluoro Methoxy) Proguanil (**4C**)

Yield: 53.5%. Mp: 197–199 °C. ^1^H NMR (600 MHz, DMSO-*d_6_*) δ 9.83 (s, 1H), 8.06 (s, 1H), 7.53–7.25 (m, 5H), 6.85 (s, 1H), 3.09 (d, *J* = 7.3 Hz, 2H), 1.47 (s, 2H), 1.31 (d, *J* = 9.2 Hz, 2H), 0.92–0.81 (m, 3H). ^13^C NMR (150 MHz, DMSO-*d_6_*) δ 160.84, 157.45, 144.78, 143.73, 123.41, 122.14, 121.95, 121.46, 119.76, 41.95, 30.46, 19.97, 14.05. HRMS (ESI) (*m*/*z*) [M+H]^+^ calcd for C_13_H_19_F_3_N_5_O, 318.1542; found, 318.1541. Purity: 97.52% (by HPLC).

##### *N*-1-Amyl-*N*-5-(4-trifluoro Methoxy) Proguanil (**5C**)

Yield: 52.9%. Mp: 196–198 °C. ^1^H NMR (600 MHz, DMSO-*d_6_*) δ 8.06 (s, 1H), 7.51–7.26 (m, 5H), 6.87 (s, 1H), 3.08 (q, *J* = 6.6 Hz, 2H), 1.53–1.38 (m, 2H), 1.33–1.18 (m, 4H), 0.85 (dq, *J* = 11.3, 6.2 Hz, 3H). ^13^C NMR (150 MHz, DMSO-*d_6_*) δ 160.72, 157.48, 143.76, 138.91, 123.15, 121.94, 121.46, 119.76, 42.21, 28.93, 28.07, 22.23, 14.31. HRMS (ESI) (*m*/*z*) [M+H]^+^ calcd for C_14_H_21_F_3_N_5_O, 332.1698; found, 332.1695. Purity: 96.17% (by HPLC).

##### *N*-1-Hexyl-*N*-5-(4-trifluoro Methoxy) Proguanil (**6C**)

Yield: 49.3%. Mp: 206–208 °C. ^1^H NMR (600 MHz, DMSO-*d_6_*) δ 9.76 (s, 1H), 7.39 (dd, *J* = 119.5, 8.6 Hz, 5H), 6.84 (s, 1H), 3.08 (q, *J* = 7.3, 6.5 Hz, 2H), 1.27 (s, 8H), 0.86 (s, 3H). 13C NMR (150 MHz, DMSO-*d_6_*) δ 160.81, 160.36, 143.75, 138.93, 122.12, 121.93, 121.46, 119.76, 118.07, 41.46, 31.36, 29.63, 26.41, 22.48, 14.36. HRMS (ESI) (*m*/*z*) [M+H]^+^ calcd for C_15_H_23_F_3_N_5_O, 346.1855; found, 346.1908. Purity: 97.94% (by HPLC).

##### *N*-1-Heptyl-*N*-5-(4-trifluoro Methoxy) Proguanil (**7C**) 

Yield: 43.5%. Mp: 210–212 °C. ^1^H NMR (600 MHz, DMSO-*d_6_*) δ 7.40 (dd, *J* = 124.3, 8.8 Hz, 5H), 6.86 (s, 1H), 3.09 (s, 2H), 1.25 (s, 10H), 0.87 (t, *J* = 6.7 Hz, 3H). ^13^C NMR (150 MHz, DMSO-*d_6_*) δ160.79, 154.14, 143.73, 138.96, 121.91, 121.46, 119.77, 42.24, 31.65, 28.80, 27.42, 26.27, 22.51, 14.40. HRMS (ESI) (*m*/*z*) [M+H]^+^ calcd for C_16_H_25_F_3_N_5_O, 360.2011; found, 360.2016. Purity: 98.30% (by HPLC).

##### *N*-1-Octyl-*N*-5-(4-trifluoro Methoxy) Proguanil (**8C**)

Yield: 45.7%. Mp: 214–216 °C. ^1^H NMR (600 MHz, DMSO-*d_6_*) δ 9.80 (s, 1H), 7.51–7.23 (m, 5H), 6.83 (s, 1H), 3.08–3.03 (m, 2H), 1.34 (d, *J* = 132.7 Hz, 12H), 0.86–0.81 (m, 3H). ^13^C NMR (150 MHz, DMSO-*d_6_*) δ 160.77, 154.13, 143.74, 138.95, 122.12, 122.04, 121.91, 121.46, 119.77, 42.22, 31.71, 29.67, 29.09, 28.40, 26.76, 22.54, 14.40. HRMS (ESI) (*m*/*z*) [M+H]^+^ calcd for C_17_H_27_F_3_N_5_O, 374.2168; found, 374.2185. Purity: 99.70% (by HPLC).

##### *N*-1-Nonyl-*N*-5-(4-trifluoro Methoxy) Proguanil (**9C**)

Yield: 42.9%. Mp: 213–215 °C. ^1^H NMR (600 MHz, DMSO-*d_6_*) δ 7.74–7.05 (m, 4H), 6.84 (s, 1H), 3.19–3.02 (m, 2H), 1.24 (s, 14H), 0.84 (tq, *J* = 8.5, 3.8, 3.2 Hz, 3H). ^13^C NMR (150 MHz, DMSO-*d_6_*) δ 160.29, 153.80, 144.13, 135.66, 123.08, 121.86, 119.68, 49.06, 31.70, 29.72, 29.35, 29.11, 29.10, 22.92, 22.53, 14.38. HRMS (ESI) (*m*/*z*) [M+H]^+^ calcd for C_18_H_29_F_3_N_5_O, 388.2324; found, 388.2326. Purity: 100.00% (by HPLC).

##### *N*-1-Decyl-*N*-5-(4-trifluoro Methoxy) Proguanil (**10C**)

Yield: 41.2%. Mp: 216–218 °C. ^1^H NMR (600 MHz, DMSO-*d_6_*) δ 9.78 (s, 1H), 7.59–7.09 (m, 5H), 6.86 (s, 1H), 3.14 (q, *J* = 7.3 Hz, 2H), 2.75 (t, *J* = 7.6 Hz, 2H), 1.58–1.02 (m, 14H), 0.87 (t, *J* = 6.8 Hz, 3H). ^13^C NMR (150 MHz, DMSO-*d_6_*) δ 160.39, 155.89, 143.64, 137.72, 121.74, 119.77, 118.07, 46.00, 31.76, 29.37, 29.33, 29.16, 29.05, 27.68, 26.38, 22.56, 14.40. HRMS (ESI) (*m*/*z*) [M+H]^+^ calcd for C_19_H_31_F_3_N_5_O, 402.2481; found, 402.2482. Purity: 99.53% (by HPLC).

##### *N*-1-Dodecyl-*N*-5-(4-trifluoro Methoxy) Proguanil (**12C**)

Yield: 40.8%. Mp: 225–227 °C. ^1^H NMR (600 MHz, DMSO-*d_6_*) δ 9.06 (s, 1H), 7.38–6.67 (m, 4H), 4.79 (s, 1H), 4.04 (q, *J* = 7.1 Hz, 2H), 2.00 (s, 2H), 1.42–1.22 (m, 18H), 1.18 (t, *J* = 7.1 Hz, 3H). ^13^C NMR (150 MHz, DMSO-*d_6_*) δ 153.82, 147.55, 147.35, 138.99, 124.84, 124.61, 123.63, 123.18, 119.48, 60.23, 34.93, 34.77, 32.01, 31.76, 31.66, 30.82, 30.61, 29.89, 29.48, 22.57, 21.23, 14.55. HRMS (ESI) (*m*/*z*) [M+H]^+^ calcd for C_21_H_35_F_3_N_5_O, 430.2794; found, 430.2825. Purity: 98.76% (by HPLC).

### 4.2. Biological Evaluation

#### 4.2.1. Reagents

Each new compound was dissolved in DMSO to prepare a stock solution of 20 mmol/L. Antibodies for the protein characterization against phosphor-AMPK (Thr172), phosphor-mTOR (S2448), phosphor-p70S6 kinase (Thr389), phosphor-4EBP1 (Thr-70) and β-actin were purchased from Cell Signaling Technology.

#### 4.2.2. Cell Lines and Culture Conditions

Three human bladder cancer cell lines (T24, UMUC3, J82) were obtained from Dr. Guo Peng (Xi’an Jiaotong University, Xi’an, China). The human ovarian cancer cell line (SKOV3) was gifted by Dr. Zhang Yong from Xiangya Hospital, Changsha, China. The human ovarian cancer cell line (OVCAR3) was purchased from ATCC. All cells were cultured in DMEM (Hyclone, Logan, UT, USA), MEM (Hyclone, Logan, UT, USA) or RPMI-1640 (Hyclone, Logan, UT, USA) medium, and the medium was supplemented with 10% fetal bovine serum (Hyclone, Logan, UT, USA) and 1% penicillin/streptomycin mixture. The cells were cultured in a 37 °C, 5% CO_2_ humidified incubator. 

#### 4.2.3. MTT Cell Viability Assay

The cell seed plate was cultured overnight in a 96 well plates (6000 cells/well), and then different concentrations of compounds were added and treated for 72 h. A total of 50 μL of 2 mg/mL MTT was added to each well, incubated for 6 h in an incubator, removed the medium, 150 μL of DMSO was added to each well, and then the absorbance of each well at 490 nm was measured with a microplate reader (Biotek, SYNERGY HTX, Winooski, VT, USA). The absorbance of the cells without drug treatment was set to 100%, the relative survival rate of the cells was calculated, and the IC_50_ value was calculated using SPSS.

#### 4.2.4. Clonogenic Assay

The cells were seeded in a 24 well plates at a density of 1000 cells/well. After 24 h, different concentrations of compounds were added and treated for 6–8 days. Cells were fixed with 10% paraformaldehyde, stained with 0.1% crystal violet, and the absorbance of each well was measured by scanning the area at 550 nm with a microplate reader (Biotek, SYNERGY HTX, Winooski, VT, USA).

#### 4.2.5. Western Blot Analysis

The cells are extracted from the protein extract solution, separated by sodium dodecyl sulfate polyacrylamide gel electrophoresis, and then transferred to a polyvinylidene fluoride (PVDF) membrane. They were blocked with 5% milk, placed the membrane at 4 °C to bind with the specific primary antibody for 15–18 h and then, rinsed with phosphate buffer containing 0.1% Tween; the secondary antibody was incubated for 1 h, and then rinsed with PBST. The Pierce super signal chemiluminescence substrate was added, and immediately the Chemi Doc system was used for blot imaging (Bio-Rad, Hercules, CA, USA). ImageJ software was used to perform grayscale analysis, and compared with the control group, and the expression level of related proteins was normalized through the expression of the internal reference protein. 

#### 4.2.6. Statistical Analyses

All data were statistically analyzed using the Bonferroni test function of SPSS software and GraphPad Prism software. The two groups of experiments used Student’s *t*-test, and the two or more groups used one-way analysis of variance to evaluate the significance of differences between the groups. The graph was generated using GraphPad Prism 6.0 software. * *p* < 0.05, ** *p* < 0.01, and *** *p* < 0.001. 

## 5. Conclusions

In summary, we designed and synthesized 10 biguanide compounds containing trifluoromethoxy group, fixed the trifluoromethoxy group on one side of the benzene ring of the biguanide group and changed the side chain on the other side. The anti-proliferative activities of the synthesized compounds were tested; we found that there is an optimized carbon chain length in the range from **6C** to **8C** with maximum anti-proliferative activities. In order to further explore the anti-tumor mechanism of the compound, Western blot results show that compound **8C** (n-octyl) has a dramatic effect on the AMPK-mTOR signal pathway. Therefore, on the basis of this study, we came to a conclusion that for the structural modification of biguanide derivatives, when the carbon chain is in the range of n-amyl to n-octyl, the biguanide derivatives have maximum anti-proliferative activities, which will guide the subsequent development of biguanide compounds in future. In meantime, we believe that **8C** is a promising anti-cancer drug targeting the AMPK pathway, which is worthy of further study.

## Data Availability

The data presented in this study are available in Appendix A.

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
