# Peer review of "Synthesis, Anti-Proliferative Evaluation and Mechanism of 4-Trifluoro Methoxy Proguanil Derivatives with Various Carbon Chain Length"

_molecules, 2021, doi:10.3390/molecules26195775_

Round 1

Reviewer 1 Report

The article is devoted to the synthesis and study of the anticancer activity of new 4-trifluoro methoxy proguanil derivatives. The new compounds have shown activity against several cancer cell lines. The authors found some patterns "structure-biological activity". The manuscript is generally of high quality. The biological part is excellent.  The novel compounds have a great potential for developing new anti-cancer candidates. There are some comments to the authors.
1. The synthesis and yields of compounds are not discussed in any way. Have the authors tried to optimize synthetic pathways? 
2. There are no alkylamines in the scheme 1.
3. There are some typos: extra spaces on lines 71, 119.
4. Abbreviated journal name must be in italics and the year in bold in references.
In general, the manuscript is suitable for publication in Molecules.

Author Response

Thank you very much for your reviewing our manuscript. Our responses to your questions are as follows:

  1. There are four synthesis routes available to our best knowledge. We chose the current route according the yield and feasibility. For this rational, we added this information in Discussion Section.
  2. Thank you very much for your careful check. Alkyl amine was added in step b in Scheme 1.
  3. Thanks a lot for your important suggestions. The typos were carefully examined and corrected.
  4. Thank you very much for your points. The format of the references was adjusted.

Reviewer 2 Report

Due to its inert property, trifluoromethyl/trifluoromethoxy is of great interest and frequently applied to the medicinal chemistry perspective. In this submitted manuscript titled “Synthesis, anti-proliferative evaluation and mechanism of 4-trifluoro methoxy proguanil derivatives with various carbon chain length”, the authors presented the synthesis of ten new trifluoromethoxy-containing proguanil derivatives by installing various linear carbon chains, evaluated their antiproliferative activities, and identified the potential pharmacological mechanism. As a result, the authors discovered derivatives with a moderate length of linear alkyl chain (n=5-8) exhibiting the best antiproliferative activities in several cancer cell lines and halting colony formation. Furthermore, the authors revealed that these potent derivatives regulate cell proliferation by modulating the AMPK/mTOR pathway.

I find this submitted manuscript intriguing and insightful for future medicinal chemistry programs of developing proguanil-related chemical scaffolds as anticancer agents. However, regretfully the submitted manuscript should not be considered at this moment due to several serious flaws:

Flaws:

1) Please make sure the final products are right:

a): the proton and carbon NMRs are not right, at least for compounds 2C, 10C, 12C

2) Please make sure final products are pure enough for biological assays, at least for compounds 4C, 5C, 9C, the purities are of great concern.

3) HPLC data for compounds do not make sense: the retention times are not corresponding well to the lipophilicity changes.

4) Controls are missing for all the colony formation assays

Author Response

Thank you very much for your reviewing our manuscript. Our responses to your questions are as follows:

  1. The characterization assays of all compounds were either re-performed or re-analyzed to confirm that their structures are correct and pure enough. The NMR of 2C, 7C, 9C and 12C, and HPLC of all compounds using brand new column (Agilent 5 TC-C18(2) 250×6 mm) were re-performed and re-analyzed these updated data. All spectra are attached in supporting information.
  2. Thank you very much for your expertise concerns. We combined the processed data of NMR by using MestReNova software and re-performed HPLC results of all compounds as stated above to confirm the purity of these compounds in the revised submission.
  3. Thank you very much for your great points. When we performed HPLC in the first submission, we only focused on purity but did not pay attention on retention time. After you raised the question, we re-performed HPLC as stated above and found that the retention time was consistent with lipophilicity changes. These newly collected data is presented in supporting information.
  4. Thank you very much for your excellent suggestions. For this study, the appropriate control is Proguanil. Thus, we did extra experiment to detect the effect of proguanil on the colony formation of J82 cell line. The results are presented in supporting Information. Furthermore, the effects of Proguanil on colony formation of T24 and UMUC3 cell lines have been reported in our previous studies, which was shown in reference 17 in the revised manuscript. All these information was added in the revised manuscript.

Round 2

Reviewer 2 Report

Due to its inert property, trifluoromethyl/trifluoromethoxy is of great interest and frequently applied to the medicinal chemistry perspective. In this submitted manuscript titled “Synthesis, anti-proliferative evaluation and mechanism of 4-trifluoro methoxy proguanil derivatives with various carbon chain length”, the authors presented the synthesis of ten new trifluoromethoxy-containing proguanil derivatives by installing various linear carbon chains, evaluated their antiproliferative activities, and identified the potential pharmacological mechanism. As a result, the authors discovered derivatives with a moderate length of linear alkyl chain (n=5-8) exhibiting the best antiproliferative activities in several cancer cell lines and halting colony formation. Furthermore, the authors revealed that these potent derivatives regulate cell proliferation by modulating the AMPK/mTOR pathway.

Again, I find this submitted manuscript intriguing and insightful for future medicinal chemistry programs of developing proguanil-related chemical scaffolds as anticancer agents. In this revised submission, the authors have successfully addressed several serious flaws I came up with previously, however, prior to its acceptance by Molecules, I recommend the authors further look at the following points:

Major points:

1)In the colony formation assay, the authors should note that their tested concentrations of the selected compounds are questionable as inhibition of the colony growth is not very convincing. I suggest the authors should test higher concentrations (next to IC50s) to see nearly complete colony growth inhibition.

2)In the SI, signal/noise ratio for many 13C-NMR spectra are not apparent enough, for example: 2C, 7C. Longer acquisition time are recommended.

3) In the SI, C-1 on the phenyl ring are not show in the 13C-NMR spectra for compound 3C

4) In the SI, structures of the following final products are not supported by the NMR spectra:

a) 12C: proton signals of C-13 are not present in the 1H-NMR

b) 10C: what is the signal at δ 2.75 ppm in the 1H-NMR? What are the signal at δ174 ppm in the 13C-NMR

c) 9C: why are signals of the alkyl chain not present in the 13C-NMR? They should be easier to shown than other carbons

Minor points:

1) In the result part, synthesis of the target compounds should be briefly illustrated although the chemistry is well-known.

2) In Scheme 1: carbon numbers of compounds 10C and 12C are wrong; also compound 7C is missing.

3) In Table 1:

a) unit (uM) of the IC50 should be noted;

b) SD or SEM should be given for IC50s.

c) How many replicated were done for IC50s?

4) Size of the chemical structure should be consistent. Some of the structures are squeezed.

5) In Figure 6, how long are the cell treated? Also, how are the cells treated in the WB (like cell numbers, time point)?

6) In the chemistry pat:

a) please kindly explain what are the ppm, Hz.

b) Detailed information of HPLC instrument, column, solvent system.

7) What are the physical properties of the final product? If solid, melting point should be given.

8) HRMS data for 9C, 10C, 12C are missing (What you provide are low resolution data)

9) Line 253 should be RPMI-1640.

10) Reference: title of the articles are not given, please format correspondingly.

Author Response

Thank you very much for your reviewing our manuscript. Our responses to your questions are as follows:

For major points:

  1. We tested the inhibitory capacities of these compounds to colony growth at higher concentrations and have updated the results into the second revised manuscript.
  2. Thank you very much for your suggestion, the 13C-NMR of 2C and 7C were re-performed with longer acquisition time, further confirming the chemical structures of these two compounds with better ratio of signaling to noise. All updated spectra are attached in supplemental information.
  3. Thank you a lot for your point, the C-1 on the phenyl ring of compound 3C is the signal at δ 143.61 ppm in the 13C-NMR.
  4. Thank you very much for your careful examination. a. The proton signal of C-13 of compound 12C is the signal at δ 4.04 ppm in the 1H-NMR. b. 10C: the proton signal of C-14 is the signal at δ 2.75 ppm in the 1H-NMR. Carbonyl carbon signal of solvent ethyl acetate is located at δ174 ppm in the 13C-NMR. c. The 13C-NMR of 9C was re-performed and re-analyzed these updated data, confirming that the chemical structure is correct. All spectra are attached in the supplemental information.

For minor points:

  1. Thank you very much for your suggestions, we have supplemented the process of target compound synthesis in result section.
  2. Thank you very much for your careful check, we have corrected Scheme 1.
  3. Thank you a lot for your important points, we have revised and supplemented the Table1 in the second revised manuscript.
  4. Thank you very much for your check, we have adjusted the size of the compound structure to keep them consistent in the second revised submission.
  5. Thank you very much for your helpful comments, we have added this information in the second revised manuscript.
  6. Thank you very much for your points. Hertz, in short Hz, is the basic unit of frequency. ppm stands for Parts Per Million and it represents frequency stability of oscillators and crystals. Δf(Hz)/f(Hz) = FS(ppm) / 1000000. b. Thank you very much for your suggestion, we have added the detailed information of HPLC instrument, column, solvent system in the second revised manuscript.
  7. Thank you very much for your constructive comments, we have added the melting points of all compounds in the second revised manuscript.
  8. Thank you a lot for your points. Since we were interested in compounds with good anticancer activity, which we did perform their high-resolution mass spectrometry. Since their anticancer activities of compounds 9C, 10C and 12C were not great, thus we only did perform low resolution mass spectrometry. However, all data from the 1HNMR ,13CNMR and low-resolution MS confirmed their chemical structures.
  9. Thank you very much for your careful check, the typos were carefully examined and corrected.
  10. Thank you very much for your suggestions, the format of the references was adjusted.